

# A system for analysis of H$_2$ and Ne in polar ice core samples

Eric S. Saltzman[1,2], Miranda H. Miranda[1], John D. Patterson[1], and Murat Aydin[1]

[1]Department of Earth System Science, University of California, Irvine, Irvine, CA, USA 92697-3100
[2]Department of Chemistry, University of California, Irvine, Irvine, CA, USA 92697-3100

*Correspondence to*: Eric S. Saltzman (esaltzma@email.com)

**Abstract.** This paper describes instrumentation and procedures developed to measure H$_2$ and Ne in polar ice core samples. Gases are extracted from ice core samples by melting under vacuum. Measurements are conducted by gas chromatographic separation with detection by a pulsed helium ionization detector (He-PPD). The analytical system was developed for field analysis of ice core samples immediately after drilling. This minimizes the potential for exchange of these highly permeable
gases between the ice core and the modern atmosphere. The design, operation, and performance of the instrument are discussed using data from the initial deployment to Summit, Greenland. The results demonstrate the feasibility of ice core analysis of H$_2$ and Ne with precision of 8.6% and 10.2% (1σ) respectively.

Short Summary: This study describes a system for analysis of hydrogen (H$_2$) and neon (Ne) in polar ice core samples in the
field immediately after drilling. The motivation is to reconstruct the atmospheric history of H$_2$ to improve understanding of global H$_2$ biogeochemistry and how it has varied over time. This knowledge will help inform models used to project future atmospheric levels of H$_2$ and assess the climate impacts of widespread utilization of H$_2$ as an energy source.

## 1 Introduction

The global biogeochemical cycle of atmospheric H$_2$ is complex, involving both natural and anthropogenic sources and losses to soils and reaction with atmospheric OH (Novelli et al., 1999; Paulot et al., 2021). Research on the biogeochemical cycle of H$_2$ has been stimulated by efforts to expand the H$_2$ energy sector, which would likely result in increased emissions. Efforts are ongoing to quantify the impact of increased emissions on air quality and climate (Warwick et al., 2022, 2023; Derwent et al., 2020). Reconstructing the paleo-atmospheric history of H$_2$ provides an opportunity to assess both natural
variability in sources/sinks and anthropogenic impact on in the global H$_2$ budget. Firn air studies show that atmospheric H$_2$ levels increased by about 65% over the past century (Patterson et al., 2023).
Polar ice core records could extend these records across the entire industrial era and capture a wider range of response of atmospheric H$_2$ to climate variability. At present, there are no paleo-atmospheric records of H$_2$ recovered from ice cores. This is largely due to the high permeability of H$_2$ in ice which causes ice core samples to rapidly equilibrate with the modern
atmosphere after drilling (Haan, 1998; Patterson & Saltzman, 2021). The mobility of H$_2$ in ice is a result of its small



molecular diameter (2.89 Å), which is small relative to the 4.5 Å gap in the hexagonal structure of ice 1H. Neon has a similar atomic diameter (2.75 Å) and similar permeability in ice. The mobility of $H_2$ and Ne in ice also leads to pore close-off fractionation during the air entrapment process (Patterson et al., 2020, 2021, 2023; Severinghaus & Battle, 2006). Measurements of Ne in ice cores may be a useful diagnostic tool for quantifying the effects of pore close-off fractionation on the levels of $H_2$ in polar firn and ice (Patterson et al., 2020).


Here we describe instrumentation for the analysis of $H_2$ and Ne in polar ice core samples. The instrument was developed to extract and analyse samples in the field immediately after they are recovered from the ice core drill, with the goals of 1) determining *in situ* $H_2$ and Ne levels, and 2) determining the rate of equilibration of ice core samples with modern air. The instrumentation was deployed for the first time at Summit, Greenland during the summer of 2024. The design, operation,


and performance characteristics of the instrument are discussed and some results from the initial deployment are presented. The instrument described here consists of: 1) a vacuum system for extracting air from ice core samples and transferring the air to a GC injection loop or to a flask for storage, and 2) a gas chromatograph, detector, and data acquisition electronics. The system is based on commercially available components, with the exception of a custom-made piston pump for transferring and pressurizing small air samples into an injection loop to improve detection sensitivity. There are surprisingly


few off-the-shelf solutions to this problem described in the analytical literature. The construction and operation of the piston pump is therefore described in detail.

## 2 Instrumentation

### 2.1 Ice core extraction line

The ice core extraction line (Fig. 1) is a stainless-steel vacuum system consisting of an oil-free vacuum pump (ScrollLabs


model SVF-E0-5PFS), a capacitance manometer (MKS Baratron), welded bellows valves and ball valves (Swagelok models SS-4H and SS-41GXS2). The manifold is constructed largely with 1/8" stainless-steel tubing and Swagelok fittings. A bulkheaded ¼" ultra-torr adapter and short length of ¼" PFA (or Synflex 1300) tubing serves as the connection to the glass ice core melting chamber. The manifold also includes a Nafion membrane tube inside a box filled with Drierite dessicant. The extraction line was designed to transfer air extracted from an ice core sample to the GC injection loop and/or to a glass


sample flask for later analysis.



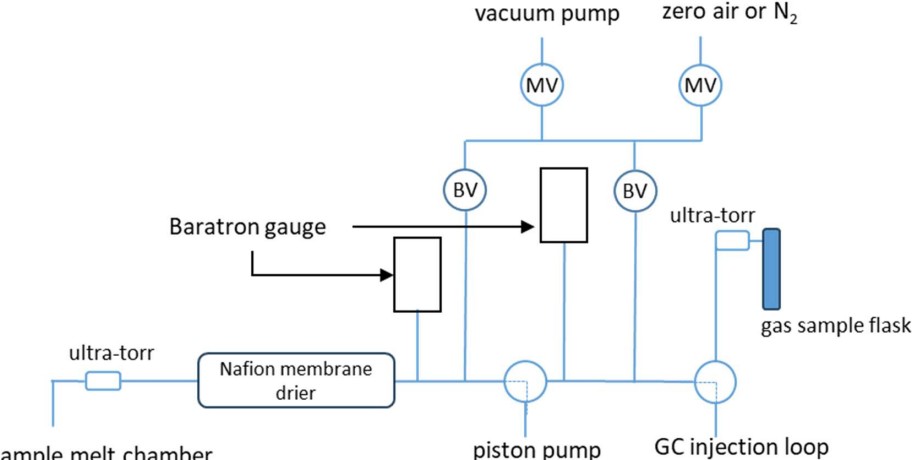

**Figure 1: Schematic of extraction line used for the measurement of H₂ and Ne in polar ice cores. Circles labelled "BV" represent bellows valves, and the unlabeled circles with dotted lines represent 3-way ball valves.**

## 2.2 Ice core extraction chamber

Ice core samples are melted in a 500 cm$^3$ glass chamber (Fig. 2). The lower portion of the chamber is a flask constructed from one side of a 75 mm borosilicate glass O-ring joint (Ace Glass, 7646-18) with the tube end closed. The surface of the O-ring joint is ground flat. The top of the chamber is of a ¼" flat borosilicate glass plate with a glass valve (Glass Expansion K2B3-08) fused at the center of the plate. The chamber is sealed by an O-ring (Viton A, size #239) located between two concentric plastic rings. These rings serve as a removable "groove" to locate the O-ring, limit compression when the chamber is evacuated, and prevent liquid water from contacting the O-ring. The outer ring is held in place by a ¼" lip that fits over the outer wall of the O-ring joint. This piece is external to the chamber and does not contact the sample. These rings are 3D printed using polylactic acid (PLA). The inner ring which contacts the ice core sample is machined from 1/8" thick PTFE sheet.



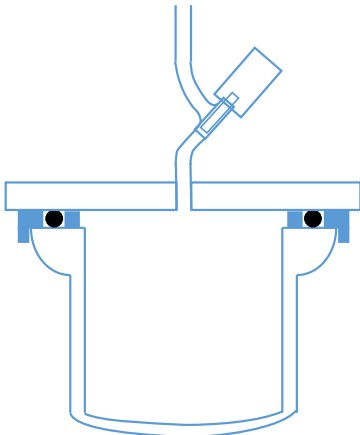

**Figure 2: Cross -section of the 500 cm³ glass extraction chamber used for the measurement of $H_2$ and Ne in polar ice cores.  The valve is a glass gas expansion valve with PTFE O-rings.  Black circles represent the Viton O-ring and the solid shapes represent concentric plastic rings that form a removable O-ring groove.**

**2.3 Sample transfer piston pump**

A sample transfer piston pump was developed and fabricated in-house to deliver ice core air to the injection loop of the gas

chromatograph or to a glass flask for storage (Fig. 3).  The pump cylinder was made from a precision bore borosilicate glass tube (1" ID, 20 cm long; Ace Glass), with a narrow bore ¼" OD glass tube fused to one end as the inlet.  The other end of the tube was slightly flared to facilitate insertion of the piston.  The piston was machined in four parts from solid rod (316 stainless-steel, PEEK, or MACOR machinable ceramic) and assembled with stainless-steel bolts.  The multi-segment design of the piston allowed the piston seals to be installed without distortion.  When assembled, the piston has three square

grooves, each 1/8" wide and 1/16" deep with 0.03" radiused outer edges to avoid damaging the O-rings during assembly. The piston was supported at two grooves by wear rings made from split PTFE washers (0.750" ID, 1.000" OD, McMaster-Carr 95630A251).  The third groove contains the dynamic seal which is in contact with the sample air.  The piston has a #8 blind threaded hole in one end that is used to connect the piston to the linear actuator arm.  In the expanded position, the cylinder volume in contact with the sample air has a volume of approximately 100 cm³.  The piston has a travel distance of

20 cm and stops within 1 mm of the cylinder end.  When compressed, the volume of the cylinder and end tube is approximately 2 cm³, giving a compression ratio of approximately 50:1.  There is additional volume associated with tubing and fittings between the pump and the injection loop, so the actual compression ratio is considerably lower (roughly 10:1). The requirements for the piston seal are:  1) a leakage rate of ambient air into the vacuum system that is small relative to the gas sample, 2) negligible outgassing of detectable $H_2$ or other gases of interest, and 3) reasonably long service lifetime

relative to the duration of a field project, and 4) tolerance for temperature variability typically encountered in field





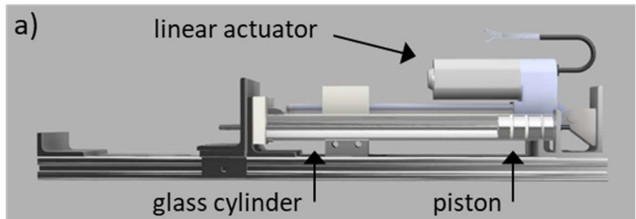

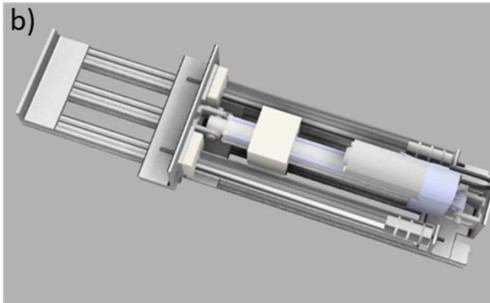
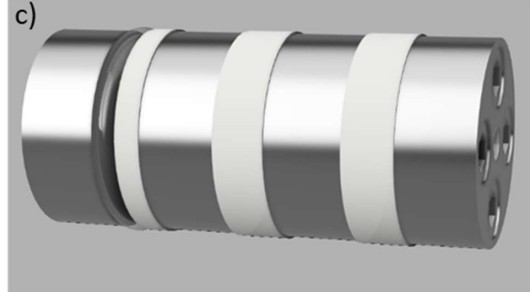

**Figure 3: Dual cylinder piston pump for gas sample compression: a) side view, b) top view, c) expanded view of piston.**

conditions.  A Teflon (PFA or FEP) encapsulated Viton A O-ring was used as the piston seal (#210, I.D. 0.734", cross section 0.139", Row Inc.).  Encapsulated O-rings are not typically recommended for use as dynamic piston seals, presumably because of rapid wearing of the sealing surface under the high shear and high temperatures often associated with rapid piston cycling in industrial applications.  In this low-speed application wear rates were not a determinative factor.  These seals

required replacement in the field, most likely from exposure to wide ranging temperatures.  Under laboratory conditions, seals in regular use require replacement every 1-2 months.

Several other types of seals were tested for this application.  Stainless-steel spring-energized PTFE seals exhibited excellent mechanical stability but an unacceptably high air leakage rate (McMaster-Carr, 5383N18).  Viton O-rings were tested with and without vacuum baking and/or lubrication (Krytox LVP fluorocarbon grease).  These maintained an excellent static

vacuum seal but exhibited $H_2$ outgassing after exposure to air.  This outgassing associated with movement of the O-ring.  It seems likely that the outgassing is related to expulsion of gases trapped in voids in the polymer due compressive or shear forces during use.  This $H_2$ likely originates from chemical reactions in the Viton rather than from permeation of ambient air because O-ring did not outgas detectable Ne or $CH_4$.  Permeation rates of Ne and $H_2$ in polymers are relatively similar due to their similar atomic/molecular size (Brehm et al., 1987).  We also tested a lubricating fluorocarbon coating (Cytonix,

FluoroSyl 880) deposited from a suspension in methanol onto the glass cylinder inner surface and heat bonded. No combination of seal and lubricant outperformed the Teflon encapsulated Viton O-ring with no lubrication.

The piston pump was driven by an electromechanical linear actuator with DC motor and 8" stroke.  Feedback between motor and piston is required to maintain a constant piston speed because the pressure differential between pump cylinder and





surrounding atmosphere varies dramatically during the analysis. Two commonly used types of sensors were tested for this

application: 1) resistance of a 3-wire 10 kΩ potentiometer providing positional feedback (Firgelli FA-PO-150-12V-8) and 2) Hall sensor providing pulses with a frequency proportional to actuator speed (Firgelli F-SD-H-220-12V-8). Both types of feedback were successfully implemented for use in this application. The advantages/disadvantages of the two approaches are discussed briefly below.

The pump operation is controlled by an Arduino Mega 2560 microprocessor operating as a PID

(proportional/integral/derivative) controller. The Arduino controller input is obtained from the feedback sensor of the linear actuator. For the positional feedback method, a high precision 5V reference voltage (AD584) was supplied to the potentiometer and to the Arduino analog to digital converter. For the Hall sensor, the pulse edges were detected by an Arduino digital input configured as a software interrupt and the microprocessor clock was used to determine pulse frequency. The linear actuator motor was driven by a pulse width modulated (PWM) Arduino output with the base

frequency adjusted to 490 Hz to reduce audible noise. This control signal was applied to a L298N dual H-bridge PWM motor controller to power the DC motor of the linear actuator (BTS7890).

The PID loop is designed to maintain constant piston speed. For the potentiometer equipped actuator, speed is determined from the difference in position between iterations of a software loop. For the Hall sensor, pulse frequency provides a direct measurement of piston speed. The set point for piston speed is nominally 1 stroke per minute or 3.3 mm/s and can be varied

over a wide range if needed. The control loop typically maintains speed to within a few percent of the set point.

For this project, a dual cylinder piston pump was constructed using a single linear actuator (Fig. 3). In this configuration, retraction of the linear actuator causes compression in both pump cylinders simultaneously. The two cylinders are connected in parallel providing about 200 cm$^3$ of volume for expansion of the air sample. The glass cylinder is mounted securely between two aluminium angles with the open end located in a Delrin block and the inlet/outlet end cushioned by a silicon

rubber pad. The pump frame consists of three parallel aluminium t-slot rails spaced 1" apart and 60 cm in length (80/20 Inc, 1010). The linear actuator is mounted on the central rail and glass cylinders are mounted on the outboard rails on either side. The linear actuator rod is bolted to an aluminium angle that slides along the outboard t-slot rails on two linear bearings (80/20 Inc., 6425). Stainless-steel threaded rods (1/4-20) couple the two pistons to the sliding aluminium angle. These rods are connected to the pistons using a short length of nylon or PEEK 10-32 threaded rod. The flexibility of the plastic rod

allows for minor misalignment between the actuator rod and the cylinder.

During development of the pump, a single cylinder version of the pump was constructed. The linear actuator and cylinder are mounted on a single 1" t-slot rail (80/20 Inc, 1010) 60 cm in length, with two pairs of t-slot angles mounted on the sides for mounting and stability. In the single cylinder version, the linear actuator arm is connected directly to the piston and extension of the actuator causes pump compression. The end of the linear actuator rod is drilled and tapped to accept a

flexible #8 threaded rod (PEEK or nylon) to couple the rod and piston.



**2.4 Gas chromatograph**

The field gas chromatograph is an SRI Instruments 8610C equipped with a heated 10 port valve (VICI) with a 3 cm$^3$ injection loop. Chromatographic separation was carried out using two stainless-steel 1/8" OD (.085" ID), 5m long HayesSep

DB columns packed in our laboratory (mesh size 100/120). High purity He carrier gas was passed through getters to remove traces of H$_2$ and other impurities (VICI models HP2/HPM before/after flow controllers). Three helium gas flows (carrier, backflush, and discharge) were controlled by electronic pressure control (EPC) units on the gas chromatograph. Two of the EPCs have flow restrictors downstream that came pre-installed on the 8610C system. He carrier flow was 30 cm$^3$ min$^{-1}$ with both columns in line. Detection was done using a pulsed discharge detector with a discharge gas flow of 7 cm$^3$ min$^{-1}$. (He-

PDD D2-I, VICI).

The first chromatographic column is configured as a precolumn that is backflushed midway (2.9 minutes) through the chromatographic run. This prevented water vapor, CO$_2$, and other high boiling compounds from reaching the detector. Backflushing of the precolumn was delayed until Ne and H$_2$ were detected, to avoid flow-induced disturbances to the baseline. This allowed major constituents from air (O$_2$, N$_2$, Ar) to elute from the precolumn onto the analytical column.

Allowing these high abundance gases to reach detector causes contamination that disrupts the detector baseline for several minutes. A second multi-port valve (10 port, VICI) was used as a detector bypass valve. The detector bypass valve diverts the flow effluent from the analytical column to waste for roughly 3.5 minutes during elution of the major air peaks. During this period, the discharge He flow rate is increased to compensate for the diverted column eluent.

A stand-alone He-PDD was mounted on the gas chromatograph with homemade 200V supply and temperature controller. The detector signal was amplified by the SRI gas chromatograph electrometer and A/D conversion was done by the data acquisition board in the gas chromatograph (SRI 333 A/D Board). PC-based data acquisition and control was provided using PeakSimple software (SRI). For H$_2$ and most other trace gases, the He-PDD induces photoionization of the analyte with detection of the resulting electrons. A helium plasma is generated in which photons ranging in energy from 13.5 eV to 17.7

eV are emitted from the transition of diatomic helium He2 (A$^1\sum_u^+$) to the dissociative 2He(1S$^1$) ground state (Wentworth et al., 1994). This mode of detection results in a sensitive, linear response to H$_2$ over several orders of magnitude. Neon cannot be ionized by this mechanism, as its ionization potential is 21.56 eV, which is above the He plasma photon energy. Instead, Ne is ionized in the He plasma by reaction with the He* metastable (19.8 eV). This mechanism of ionization results in a much less sensitive and non-linear response for Ne, compared to that for H$_2$ (and most other gases).

**2.5 Calibration**

Calibration is based on high pressure gas standards (1000 psi) prepared in our laboratory in stainless-steel cylinders (Swagelok 304L-HDF8-1GAL, 3.8 L). These standards contained H$_2$ (340 ppm), Ne (17.1‰), CH$_4$ (600 ppm), and CO (4 ppm) with high purity N$_2$ as the balance gas. Three high-pressure cylinders were filled prior to the field project and





intercompared prior to shipping, in the field, and after completion of the field work. The high-pressure standards are prepared by expanding the pure gases at a known pressure from a calibrated volume into an evacuated cylinder, followed by gravimetric addition of $N_2$. The accuracy and precision of preparing the standards is estimated as ±2%. The precision is confirmed by intercomparison of cylinders. The high-pressure gas standards prepared for this study are not intended for multi-year use and have not yet been intercompared with gas standards from other laboratories.

Working standards are prepared by diluting the high-pressure standards barometrically into evacuated 2L electropolished
stainless-steel flasks with bellows valves, with $N_2$ or zero air as the balance gas. Dilution was done in the field using a simple stainless-steel vacuum line with Swagelok VCR fittings and bellows valves (Swagelok, SS-4BG-V51), a heated Baratron capacitance manometer (MKS Type 631), and an oil-free vacuum pump (ScrollLabs, model SVF-E1-5) (Fig. 4). Working standards were analyzed daily and prepared fresh on a weekly basis. Comparison of working standards prepared from two different high-pressure cylinders gave no indication of drift over the course of six months (Fig. 5).


**Figure 4: Schematic of the dilution system used to produce working standards in the field. The low-pressure flask and high-pressure gas standard are connected to the line using PFA tubing. The circles marked BV indicate bellows valves.**

**2.6 Thermal considerations for field work**

One of the challenges for the deployment of the system at Summit, Greenland was the interruption of generator power at night, allowing the laboratory tent cool to ambient temperatures as low as -25ºC. This raised concern about stabilization of the analytical system each day and the potential for leaks due fluctuating temperatures. To minimize these issues, we constructed a tubular aluminum frame box with insulated walls, which was placed over the analytical system each night. The box temperature was controlled using an in-line duct fan, temperature sensor, and PID temperature controller set to



20ºC. The system was powered using a 6 kW h lithium battery pack, charged by generators during the day (Goalzero, Yeti 6000X). Gas flows, oven temperature, and detector temperature were maintained at their normal operating values overnight. The nighttime power draw was approximately 80W which was sufficient to heat the box. The in-line fan served to cool the box if needed.

**3. System procedures and performance**

The procedure for processing an ice core sample begins by: 1) cutting the sample to length (10 cm for the 70mm diameter cores used in this study), 2) mechanically scraping the ice with a scalpel to remove exterior contaminants, 3) sealing the sample in the extraction chamber and cycling the chamber through several cycles of flushing with high purity $N_2$ and evacuation to the vapor pressure the ice (about 1 mbar at -20ºC). The chamber is then sealed, and a warm water bath (40ºC)

is raised to immerse most of the chamber. The ice melting process takes 20-30 minutes, and the warm water bath is removed and replaced with an ice water bath just prior to the melting of the last piece of ice. The extraction line is evacuated and isolated from the vacuum pump, and the air sample is expanded slowly through the membrane drier and into the line and piston pump in the compressed position. The pump is slowly expanded (~1 minute), withdrawing the sample from the melt chamber into the piston. The pressure is allowed to stabilize and is recorded. The 3-way valve is then positioned to direct the

pump contents either to the GC injection valve loop or to a glass flask for storage. The piston pump is then slowly compressed (1 minute) and the pressure is again allowed to stabilize and is recorded.
For immediate GC analysis, the loop was then isolated from the piston pump and the injection valve was actuated. Sample pressures in the GC injection loop for the first analysis were typically in the range of 600-750 mbar, with the variability largely due to the amount of sample removed during cleaning. After completion of the chromatographic run, the injection

loop was evacuated and flushed with $N_2$, isolated, filled with residual sample from the piston pump, and injected. Sample pressures for the second analysis were typically 20% lower. The entire chromatographic run takes approximately 10 minutes (Fig 5).



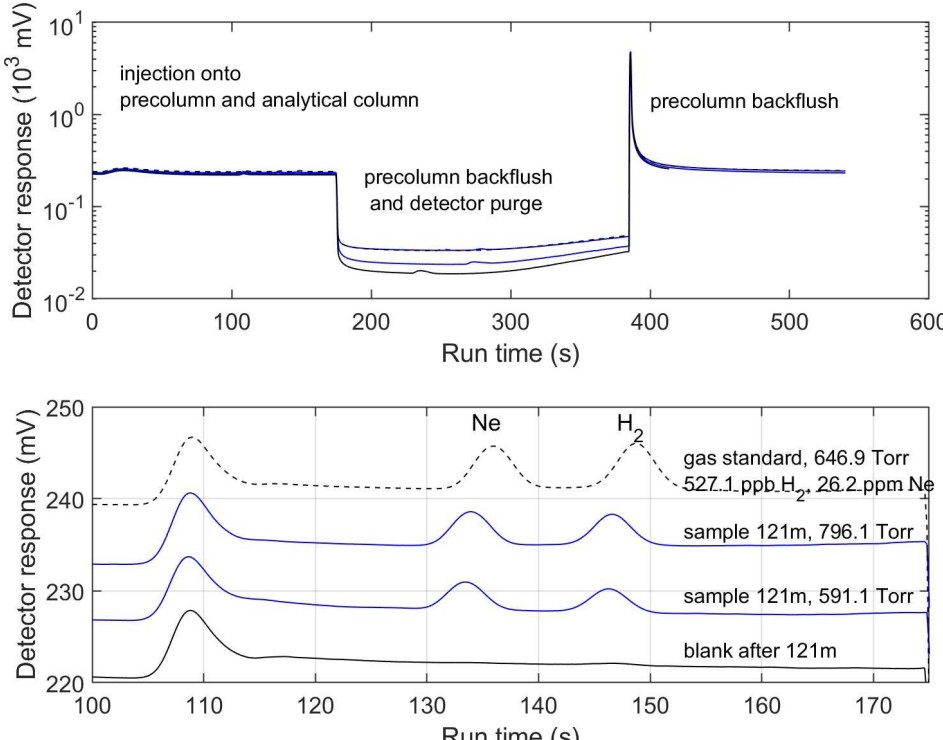

**Figure 5: Detector signals from analysis of an ice core sample (Summit, Greenland; 121m depth) and the associated blank and gas standard. Upper: Complete run from the injection, illustrating the timing of the precolumn backflush and detector purge, Lower:**
**Expanded view of the detector signals during elution of Ne and H$_2$. The early unlabelled peak is due to a change in flow rate associated with the backflush of the precolumn prior to the injection.**

Three types of blanks were assessed during the field expedition. "Pre-run" blanks involved evacuating a frozen ice core sample in the extraction flask, kept cold with a snow-filled dewar. After several flushes, the flask was filled with high-purity
N$_2$ and isolated from the vacuum line. The N$_2$ was extracted from the flask and analyzed in the same manner as a sample. "Post-run" blanks were similar, except the high-purity N$_2$ was loaded into a flask containing cold melt water after the sample had been extracted. "Standard" blanks involved loading high-purity N$_2$ directly into the piston-pump. There was not a significant difference between the three types of blanks, suggesting that the blank signal arises primarily from the piston pump itself. The three types of blanks are aggregated to apply a single blank correction to all samples and standards that
were exposed to the piston pump. The average blank H$_2$ peak area was 0.96 mV s, and the average Ne blank was 0.43 mV s, compared to typical ice core sample peak areas of 7-17 mV s for H$_2$ and 5-15 mV s for Ne. Qualitatively, blank H$_2$ peaks





appeared to be true deviations from the baseline, while the Ne peak area was more typically integration of noise on a sloping baseline (Figure 5). The $H_2$ blanks showed systematic temporal variability during the field project. We therefore fit a smoothing spline to the blank data, using the Matlab *fit* function with a smoothing parameter of $10^{-10}$. The resulting time-

dependent blank correction was applied to all samples and standards. The blank variability observed under field conditions was larger than typically encountered in the laboratory, most likely reflecting the wide variations in temperature. A recent set of laboratory measurements showed blank variability on the order of 3% of the typical $H_2$ levels in ice cores. Calibration standards at several concentrations were analyzed daily with loop pressures spanning the range encountered for the ice core samples (400-800 mbar)(Fig. 6). Standards were run through the piston pump and were blank-corrected. For $H_2$,

a preliminary calibration curve was generated from a linear regression model of the measured peak areas against the product of gas standard mixing ratio and injection loop pressure (i.e. the partial pressure of $H_2$) using the Matlab function *fitlm*. Fitting the $H_2$ standards from the entire expedition yields the calibration curve:

$$A = 0.161 + 5.98 * 10^4 * P_{H2}$$

With an $R^2$ value of 0.968 (n=158) where $A$ is integrated peak area (mV s) and $P_{H2}$ is the assigned partial pressure of $H_2$ in

the sample loop (mbar). In theory, injection loop temperature should also be accounted for in the regression, but for the Summit field deployment including temperature did not improve the regression statistics. The Ne calibration is sensitive to the total loop pressure for reasons that are not well understood but must be related to variations in detector flow rate and the mechanism of ionization (Section 2.1.3). The preliminary calibration relationship for Ne was:

$$A = -0.287 - 1.24 * 10^{-2} * P_{Loop} + 1.79 * 10^3 * P_{Ne}$$

with an $R^2$ value of 0.704 (n=158), where $P_{Loop}$ is total gas pressure in the sample loop (mbar) and $P_{Ne}$ is partial pressure of neon in the loop (mbar).

Residuals vary systematically day-to-day, indicating some sensitivity drift in the instrument. The drift is particularly acute for Ne. The magnitude of the residuals on each day scaled with peak size to some degree. Therefore, for each day, we fit the residuals with the integrated peak area as the independent variable. This daily drift correction was applied to both standards

and samples run on the same day, and then the linear regression is run a second time. The drift correction for the $H_2$ calibration is small, yielding a final calibration of:

$$A = 0.333 + 5.88 * 10^4 * P_{H2}$$

With the $R^2$ value slightly increasing to 0.985. The correction for neon is larger, with the corrected regression yielding a final calibration curve of

$$A = -1.33 - 6.80 * 10^{-3} * P_{Loop} + 1.55 * 10^3 * P_{Ne}$$

With the $R^2$ value increasing dramatically to 0.936.





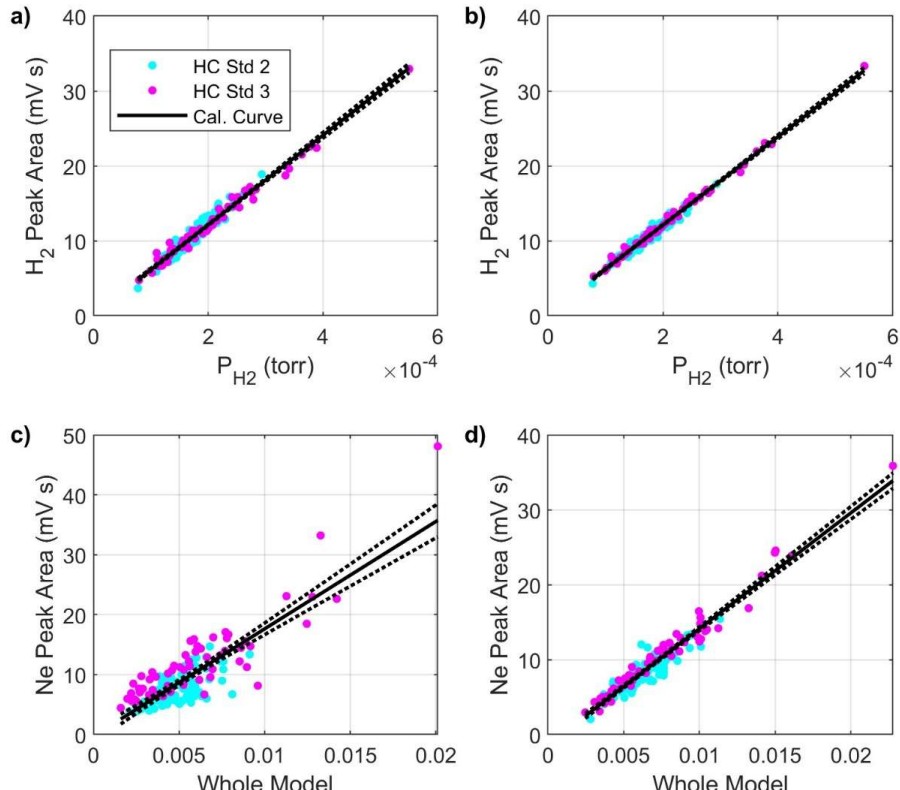

**Figure 6: Calibration data for H₂ (a and b) and Ne (c and d) obtained during the field deployment at Summit, Greenland. (a) and (c) are uncorrected for sensitivity drift. (b) and (d) are after correcting for signal drift. Peak areas are plotted against partial pressure for H₂ (a and b) because of the linearity of the calibration. For Ne, peak areas are plotted against the non=linear model described in the text (c and d). Magenta and cyan dots illustrate the comparison between data from two different high-pressure standards. See text for details.**

Mixing ratios are assigned by solving the final calibration curves for mixing ratio using the measured loop pressure. Monte

Carlo sampling is used to propagate uncertainty across the blank correction, drift correction, and calibration uncertainty.

Relative uncertainty in assigned H₂ mixing ratios average 8.6% (1σ) and relative uncertainty in assigned Ne mixing ratios

average 10.2%.

The path towards improving system precision is different for H₂ and Ne. The largest source of uncertainty in the H₂

measurements is the blank associated with the piston pump, and minor changes in the dynamic seal material or configuration



might lead to improvement. The primary source of uncertainty in the Ne measurements is sensitivity to changes in detector discharge gas and non-linearity in the detector response. For reasons discussed in Section 2.4, the detector limits the precision of our Ne measurements. Alternative detectors are needed for higher precision neon measurements.

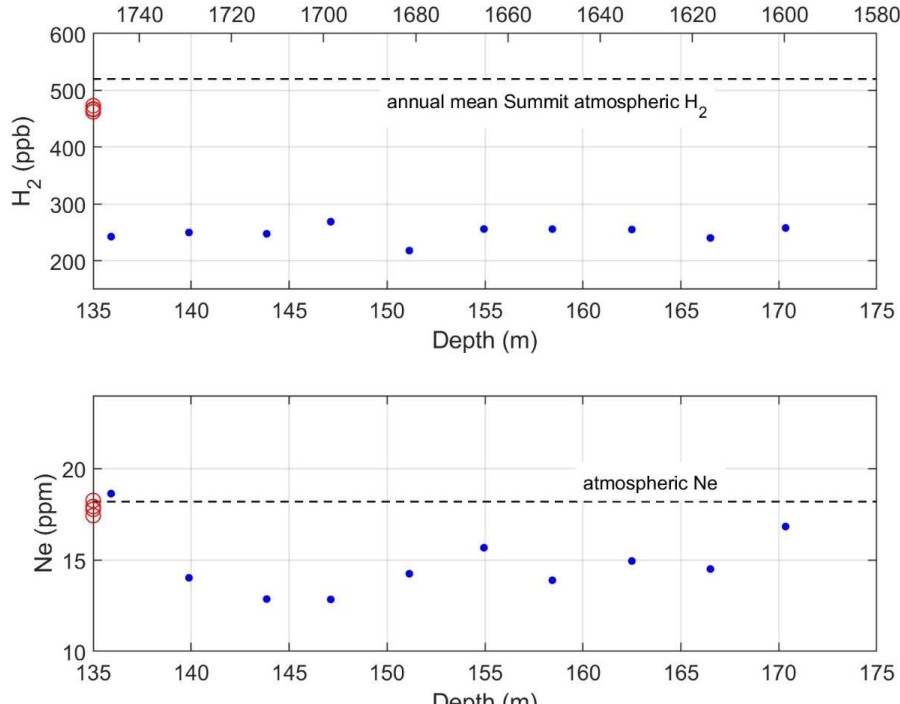

**Figure 7: Preliminary results from analysis of ice core samples from Summit, Greenland ranging from 75-150 m with approximate gas ages of 150-1750 based the age depth relationship for previous Summit ice cores (Mitchell et al.; Rhodes et al.). Upper: $H_2$ mixing ratios. The horizontal dashed line indicates annual average atmospheric $H_2$ levels at Summit from 2010-2021 from the NOAA flask network (Petron et al., 2024). Lower: Ne mixing ratios from the same ice core samples. The horizontal dashed line is the average modern atmospheric Ne level.**

Measurements of 10 ice core samples, made over 7 days, from 135-175 m depth are shown to illustrate the instrument
performance under field conditions (Figure 7; Patterson et al., submitted). The time scale assigned to these data is roughly
1600-1730 C.E., based on dating of previous Summit ice cores. For this segment $H_2$ levels were 249.2±13.6 ppb (1σ). The
variability in the data is consistent with the uncertainty in a single measurement, suggesting that atmospheric $H_2$ levels were
relatively constant over this period. For the same ice core samples, Ne levels were 14.8±1.8 ppm. Ne levels in the modern
atmosphere are 18.2 ppm. The mean Ne levels in the ice core are significantly lower. This Ne deficit is expected in ice
cores, due to the permeation of Ne out of pressurized pores during the firn air close-off process (Patterson et al., 2020;



Severinghaus & Battle 2006). Model calculations suggest that the deficit should be on the order of 3% and the deficit observed here is significantly larger. These are the first simultaneous observations of ice core $H_2$ and Ne, and this surprising result will be a focus of future study. The variance in the Ne data is consistent with the analytical uncertainty based on the calibration runs.


### 4. Conclusions

To the best of our knowledge, the system described here is the first developed to measure $H_2$ and Ne in polar ice core samples in the field in during ice coring. The system generally performed as intended and the measurements were made successfully. There is a need for further development in terms of the materials used in the piston pump dynamic seal to

reduce $H_2$ blanks and extend the working life, and a more sensitive field portable method for detection of Ne. In addition, it would be valuable to develop the capability to store ice core air extracted in the field for subsequent analysis in the laboratory. This would allow sampling on deployments where logistical limitations preclude on-site analysis, and permit higher precision analysis under laboratory conditions using a wider range of analytical instrumentation

### Author contribution

ESS, JDP, MHM designed, constructed and tested the instrumentation. MA drilled and sampled the ice core. JDP conducted the field measurements and analyzed the data. ESS prepared the manuscript with contributions from all authors.

### Competing interests

The authors declare that they have no conflict of interest.

### Acknowledgements

The authors with to thank Cyril McCormic for electronic design and troubleshooting. The authors also thank the NSF Ice Drilling Program for their support in the field through NSF Continuing Grant 2318480, field planning support from Batelle, and assistance from the NSF Ice Core Facility. This research was funded by the NSF through OPP-1907971 and by NOAA through NA23OAR4310139.



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
