# Peer review of "A system for analysis of H2 and Ne in polar ice core samples"

_EGUsphere, 2025_

## Author Comment (AC1)

Author response to Reviewer 1 (RC1) comments on:

Title: A system for analysis of H2 and Ne in polar ice core samples

Authors: Eric S. Saltzman, Miranda H. Miranda, John D. Patterson, and Murat Aydin

We appreciate the insightful comments received from the reviewer. The manuscript has improved as a result. Below are our responses to specific comments raised in the review. Excerpts from the review are shown in blue and our responses are in black. Revised manuscript text shown in italics and with quotation marks. For brevity, we include only the reviewer comments requiring a response.

I think more information could be provided about long-term performance of the system (even if only on standard runs), as well as the calibration (I cannot tell from the text, for example, what is the range of compositions of the various prepared working reference gases used to calibrate the instrument).

The performance of the system during this deployment was messy, with variability in blanks and system response much larger than typical for this system under normal conditions. This was due to a combination of harsh and variable deployment conditions in a tent and a malfunctioning electronic pressure controller on the detector discharge gas that was not repairable in the field. We thought it would be confusing for the reader to present this level of detail in the main text. As an alternative, we present the requested performance data in a Supplement with figures (S1, S2) illustrating the changes in instrument response during the field deployment

Revised text in line 210 describes the range of compositions of the working standards: "The field working standards contained concentrations ranging from 150 ppb to 500 ppb for  $H_2$  and 10 ppm to 25 ppm for Ne."

It would be helpful to see a photo of the whole instrument setup including the piston/linear actuator device.

Several photos of the system in the field and lab are included in the Supplement (Fig. S3-S5).

One of the stated goals of the instrument was to determine the rate of equilibration of ice core samples with modern air, but data toward this end are not shown. Are there preliminary results that can be included in this manuscript? If not, I recommend rephrasing this (line 42) so as not to build up the expectation for this paper.

The comment about equilibration was removed.

It would be helpful to gauge system performance if more data were shown that characterise the long-term stability of the system. Figure 6 is useful for demonstrating the linearity and gives some indication of the range of total variation, but it would be useful to see, e.g., the linearity-corrected standard data over a long period of time. I'm a little unclear how the sensitivity correction was determined. Perhaps showing the drift in a figure would help the reader gauge the timescale and magnitude of the sensitivity drift.

The sensitivity drift is shown in the Supplement figures (S1, S2). Fig. 7 (formerly Fig. 6) is the drift-corrected data.

Regarding calibration, the main text reads as if the three high-pressure cylinders had the same (or nearly the same) composition, though I think this is a mistake. Did you not measure standards of different compositions (especially standards with compositions like the ice core air) against one another to verify your calibration scheme? This seems necessary to me, especially given the lower concentrations of  $H_2$  and Ne measured in the samples.

It is correct that the three high pressure cylinders were similar in concentration. The text in section 2.5 was clarified and includes additional information about calibration standard concentrations. Line 189: "Calibration is based on three high-pressure gas standards (1000 psi) prepared in our laboratory in electropolished stainless-steel cylinders (Swagelok 304L-HDF8-1GAL, 3.8 L). These cylinders contained roughly 350 ppm  $H_2$ , 18 ppm Ne, 600 ppm  $CH_4$ , and 5 ppm CO with  $N_2$  (99.99%) as the balance gas."

The working standards used to calibrate the system covered a range of concentrations (see revised text on Line 210).

**Specific comments**

I may have just missed it, but please make sure the ice sample size is stated somewhere.

The text at Line 64 was revised as follows: "The ice cores samples in this study were 7 cm diameter, cut into 10 cm long sections for analysis (roughly 350 grams). The samples are scraped with a scalpel on all sides to clean the drilling fluid. The samples are melted in a 500 cm3 glass chamber.

Line 38 This is a bit pedantic, but I'm not sure *in situ* is exactly appropriate here given the cores are first drilled and extracted from the ice sheet prior to measurement.

Line 36 revised to: "The instrument was developed to extract and analyze samples in the field immediately after recovery from the ice core drill to avoid possible subsequent changes due to permeation during storage."

You should state what is the heating element of the oven in section 2.6. I didn't realise it was an oven until line 196.

The description of the box has been to "insulated box". The box was only heated by the GC oven. Line 221: "To minimize these issues, we constructed a tubular aluminum frame insulation box with insulated sheathing boards, which was placed over the analytical system each night."

It would be helpful for the reader to see the temporal variability in the blank to judge for themselves how significant it is. Perhaps just show the spline fit and blank data (lines 233-237).

This is now shown in the Supplement (Fig. S1).

You might consider putting the calibration equations on Figure 6 instead of listing them in the main text in lines 243-260.

We considered this but decided that the equations are more easily understood in the text where they are explained.

Line 261 – Does "calibration uncertainty" refer to the uncertainty in the standard mixing ratios? Please specify if so, and if not please also address this source of uncertainty.

Revised text in line 292: "Monte Carlo sampling is used to propagate uncertainty in the blank correction, drift correction, and standard mixing ratios. The resulting relative uncertainties in assigned mixing ratios average 8.6% ( $1\sigma$ ) for  $H_2$  and 10.2% for Ne."

Line 264 – I would change this to say, "The factors limiting system precision are different for  $H_2$  versus Ne."

**Done**

In Figure 7, please state what the red circles mean in the caption.

**Done**

Lines 267-268 There is only one sentence in section 2.4 that states the reasons for the detector limiting Ne precision (lines 168-169), so you might as well just say again here what those reasons are.

Revised text line (line 297): "The primary source of uncertainty in the Ne measurements is the generally low and non-linear response of the He-PDD detector. The mechanism of detection of Ne in the He-PDD involves reaction with metastable He\* rather than photoionization (as in the case of H2), as a consequence of the high Ne ionization energy (Sect. 2.4)."

Line 272-273 Change to something like, "..., suggesting the atmospheric H₂ levels were constant within the measurement uncertainty over this period."

Revised text line 302: "The variability in the data is consistent with the uncertainty in a single measurement, suggesting that atmospheric H2 levels were constant within the measurement uncertainty over this period."

**Technical corrections**

On the Figure 1 caption, also state what "MV" stands for. Done.

Figure 1 and Figure 2 could be combined to save space if desired. Done.

Line 99-100 Change to read, "This outgassing is associated with movement of the O-ring." Done.

Line 101 "...polymer due to compressive or shear..." Done.

Line 103 "the O-ring did not outgas..." Done.

Line 145 "This prevented(?) major constituents from air..." Done.

Line 193 "...we constructed a tubular aluminum frame oven with insulated walls..." Done.

Line 204 "...and evacuation to the vapor pressure of ice" Done.

Figure 6 caption – "non-linear" rather than "non=linear" Done.

| Figure 7 caption – references to Mitchell et al. and Rhodes et al. should have proper formatting. The age range 150-1750 is different than what is stated in the text on line 271 (1600-1730 CE). Done. |
|---------------------------------------------------------------------------------------------------------------------------------------------------------------------------------------------------------|
|                                                                                                                                                                                                         |
|                                                                                                                                                                                                         |
|                                                                                                                                                                                                         |
|                                                                                                                                                                                                         |
|                                                                                                                                                                                                         |
|                                                                                                                                                                                                         |
|                                                                                                                                                                                                         |
|                                                                                                                                                                                                         |
|                                                                                                                                                                                                         |
|                                                                                                                                                                                                         |
|                                                                                                                                                                                                         |

---

## Author Comment (AC2)

Author response to Reviewer 2 (RC2) comments on:

Title: A system for analysis of H2 and Ne in polar ice core samples

Authors: Eric S. Saltzman, Miranda H. Miranda, John D. Patterson, and Murat Aydin

We appreciate the insightful comments received from the reviewer. The manuscript has improved as a result. Below are our responses to specific comments raised in the review. Excerpts from the review are shown in blue and our responses are in black. Revised manuscript text shown in italics and with quotation marks. For brevity, we include only the reviewer comments requiring a response.

Questions mainly relate to the materials used within the setup, as Hydrogen is quite material sensitive-Line 52: It is known that Hydrogen is very material sensitive, also to aluminium, which can be exposed if a Synflex line is slightly damaged at a union for example. Further down in the text, the authors also describe how the material of the piston seal matters regarding influences on the Hydrogen concentration in the sample. In various wetted parts of the setup non-stainless-steel materials are used. Did the authors make material choices based on tests they conducted, or based on references? I think it is worth emphasizing at a suitable place in the text (possibly in the section describing the blanks) why PFA, PTFE, PEEK, etc. are readily usable; or give some references where that was shown.

We revised the text to provide additional information about how materials used in the extraction line were tested (Lines 54): "Nafion is a fluoropolymer membrane that acts as a very selective, semi-permeable membrane to water vapor. The materials used in the extraction line (stainless steel, PFA, PTFE, Nafion, Viton, Synflex 1300) were tested for  $H_2$  loss and outgassing on timescales of 1-2 hours, relevant for the extraction of the gas from the ice core."

Flow chart of the GC-PDD would be helpful to follow the descriptions in the text. Line 142: I think a flow schematic of the GC-PDD setup (including getters, flow controllers, valves, etc.) would be helpful (if there is still allowance for another display item).

We added a schematic illustrating the components of the GC-PDD system (Fig. 4)

More background information needs to be added on the preparation of standards and on the implementation of the calibration. This is necessary to fully trace the entire process of deriving the sample concentrations. At last, I was wondering whether the instrumental drift was associated with the running time of the instrument, or could be characterized in any other way?

We added details in section 2.5 to better explain the preparation of standards and how they were calibrated. Line 225 explains the running time and temperature stabilization of the instrument: "Gas flows, GC oven temperature, and detector temperature were maintained at the normal operating values overnight (~20 °C). The nighttime power usage was approximately 80W which was sufficient to heat the insulated box. The in-line fan was activated in order to cool the box if the temperature in the box reached 20 °C, which did occur during the deployment."

We did not observe measurable drift in our high-pressure standards. We believe the drift in the calibration curves resulted from instrumental drift. Line 196: "Working standards diluted from two of the high-pressure cylinders gave no indication of drift of the high-pressure cylinders over the course of six months (Fig. 5)."

Line 38: I think with "in *situ*" the authors intended to emphasize that the measurements are happening at site, so I would slightly rephrase to "determining H2 and Ne levels *in situ*".

The text was revised (Line 36) to: "The instrument was developed to extract and analyze samples in the field immediately after recovery recovered from the ice core drill to avoid possible subsequent changes due to permeation during storage."

Figure 1: Please make sure that the reader knows that "ultra-torr" means an "ultra-torr adapter", like it is phrased in the text.

Fig. 1 caption was revised accordingly.

Line 62: I have some difficulty understanding the sentence: "The lower portion of the chamber is a flask constructed from one side of a 75 mm borosilicate glass O-ring joint (Ace Glass, 7646-18) with the tube end closed." What does "from one side" and "the tube end closed" mean? Could the authors please rephrase this sentence. Or, if this is a special technical term, then it would probably help if a real picture or some labels were added to the schematic.

The text was revised as follows (Line 66): "The flask is built from a 75 mm borosilicate glass O-ring joint (Ace Glass, 7646-18). The Viton O-ring exhibited outgassing of  $H_2$  during initial tests with melting ice. This outgassing did not occur when the flask was dry, so we suspect the outgassing was related to contact of Viton with liquid water. The other side of the glass joint is glass-blown into a dome shape to close the joint and to create the lower portion of the glass chamber (Fig. 2)."

We also added additional labels to Fig. 2 to differentiate more clearly between the sample chamber top and bottom.

Line 63: If there is a specific reason for the choice of a ground flat O-ring joint, please briefly mention it.

We expanded the text in Sect. 2.2 to more clearly explain why we ground the O-ring joint.

Line 65: "limit compression" in general, or on a specific part? If meant for a specific part, please mention, e.g. "limit compression on the O-ring".

Addressed in previous comment.

Line 66: To avoid confusion between the plastic rings and the O-ring, please write "The outer plastic ring....".

Revised text (Line 74): "The outer plastic ring is held in place by a ¼" lip that fits over the outer wall of the glass joint." We also added a top view of the plastic rings to Fig. 2 for clarity.

Line 96: This may be lack of knowledge on my side, but was a standardized test developed to check on the state of the seal or on which criterium is it replaced every 1-2 months?

Revised text (Line 104): "These seals required replacement in the field, most likely from exposure to wide ranging temperatures. In laboratory conditions, the wearing of the encapsulated O-ring became noticeable in the piston "standard" blanks after 1-2 months of use. Higher  $H_2$  levels in the daily blanks were followed by the replacement of the O-ring. More information about blanks is found in Sect. 3."

Line 103: Please quickly check that when speaking of permeation rates in polymers, you really mean polymers in general, and not the polymer types that were tested in this study (as PTFE is also a polymer, but this was used in the setup).

Revised text (Line 114): "This  $H_2$  likely originates from chemical reactions in the Viton rather than from permeation of ambient air because the O-ring did not outgas detectable Ne or  $CH_4$ . Permeation rates of Ne and  $H_2$  in Viton are relatively similar due to their similar atomic/molecular size (Brehm et al., 1987)."

Line 142: Maybe change the title to something including the detector as well.

Section 2.4 title revised to: "2.4 Gas chromatograph and helium pulsed discharge detector"

Line 143: A 10 port two-position valve?

Revised text (Line 155): "The field gas chromatograph is an SRI Instruments 8610C equipped with a heated 10 port two position valve (VICI) with a 3 cm3 injection loop."

Line 145: High purity helium of grade 5 or 6?

Revised text (Line 158): "Ultra-high purity He (99.999% purity) carrier gas was passed through getters to remove traces of  $H_2$  and other impurities" (see Fig. 4.)

Line 145: How many getters of each model?

Revised text (Line 159): "Three helium gas flows (carrier, backflush, and discharge) were controlled by electronic pressure control (EPC) units on the gas chromatograph. A high-capacity getter (VICI model HP2) was used on the He supply, and three smaller capacity getters (VICI model HPM) were used in the downstream of the three electronic pressure controllers (Fig.4.)"

Line 146: Were the flow controllers already described before? Where are they located?

The electronic pressure controllers are first described in line 159 "Three helium gas flows (carrier, backflush, and discharge) were controlled by electronic pressure control (EPC) units on the gas chromatograph." The new Fig. 4 shows the flow configuration.

Line 152/ Line 154: Do I understand it right, that water vapour and  $CO_2$  do not even reach the main column, but  $O_2$ ,  $N_2$  and Ar do? If so, then please specify, e.g. "and other high boiling compounds from reaching the main column and the detector".

That is correct. We revised Line 167 to: "This prevented water vapor, CO2, and other high boiling compounds from reaching the main column and detector."

**Line 152: Was it tested how much water vapour is still left in the sample after Nafion drying? Would this have an influence on the targeted gases in the sample?**

We did not measure the water vapor levels after the Nafion but expect them to be well below 1% (molar ratio to air). The purpose is to avoid possible condensation during expansion of the sample into the sample loop and to make our samples directly comparable to dry calibration standards. The permeation of major air components ( $N_2$ ,  $O_2$ , Ar) through the membrane is undetectable in our system and we have not observed any effect on calibration standards containing  $H_2$  and  $N_2$ .

line 156: This is not a two-position valve, correct? Is this also heated? The second valve used as a bypass valve as referred to in line 171 is also a 10 port two position valve and is not heated. Revised line 171: "A second 10 port two position valve (VICI) was used as a detector bypass valve".

**Line 157: Please specify "for roughly 3.5 minutes (minute xxx-xyz through the run) during elution..."**

Both comments are addressed in revised text (Line 168): "Backflushing of the precolumn was delayed until Ne and  $H_2$  were detected to avoid flow-induced disturbances to the baseline. This allowed major constituents from air ( $O_2$ ,  $N_2$ , Ar) to elute from the precolumn onto the analytical column. Allowing these high abundance gases to reach the detector causes contamination that disrupts the detector baseline for several minutes. A second 10 port two position valve (VICI) was used as a detector bypass valve. The detector bypass valve diverts the flow effluent from the analytical column to waste. This valve is actuated for 3.5 minutes during elution of the major air peaks (minutes 3-6.5 of the run)."

**Line 159: Is it ever mentioned somewhere at what temperature the GC oven and the PDD are operated?**

Revised text (Line 156): "The heated 10 port two position valve and the chromatographic column were heated to 30 °C via the GC."

Revised text (Line 175): "A stand-alone He-PDD (VICI model D-3) was mounted on the gas chromatograph with 200V power supply. The detector temperature is maintained at 100 °C using a PID controller."

Line 160: Is the temperature controller for monitoring and setting the PDD temperature? Is it not done via the GC?

In our system, the He-PDD electronics (temperature control, electrode voltages) are stand-alone. The temperature controller sets the PDD temperature. From the GC, we only use the oven, electronic pressure controllers, electrometer, and data system.

Line 166: "orders of magnitude", including the range of atmospheric H2 abundance?

Yes. Novelli et al. (2009) demonstrated linear response of this detector to 2ppm.

Revised text (Line 179): "This mode of detection results in a linear response to  $H_2$  over several orders of magnitude, including the range of atmospheric  $H_2$  abundance (Novelli et al., 2009)."

Line 174 & Line 183: Hydrogen can drift depending on the cylinder material and cylinder batch. Was there some sort of longer-term stability testing of the used cylinders or comparison to a sample filled into

another type of cylinder to exclude drift in the cylinders? Or could you exclude simultaneous drift in the three cylinders with the comparisons you made and this question I already covered in Line 183?

Revised text (Line 181): "The high-pressure cylinders were filled prior to the field project and were intercompared prior to shipping, in the field, and after completion of the field work."

Revised text (Line 195): "Intercomparison of these standards yielded agreement within ±2% (1 $\sigma$ ) which is consistent with our estimate of the uncertainties in their preparation. The high-pressure gas standards prepared for this study are not intended for multi-year use and have not yet been intercompared with gas standards from other laboratories. Working standards diluted from two of the high-pressure cylinders gave no indication of drift of the high-pressure cylinders over the course of six months (Fig. 5)."

Line 176: Do I understand it right, that you prepared your own scale? Or is it planned to reference (at least Hydrogen) to an external scale like the WMO scale? I am not sure if an established scale exists for Neon. Is this preparation of calibration cylinders a refined procedure in your lab (e.g. at the level of a metrological institute) and is there a literature reference that describes the details? What was taken into account to derive the 2% uncertainty? I think this section either needs referencing or should be described in a bit more detail.

Yes, we did prepare our own calibration scale for this project. We have been preparing high pressure gas standards in our laboratory at UCI for numerous trace gases for more than 20 years and have participated in formal and informal interlaboratory intercomparisons. We do plan to intercalibrate for H2 with an external scale like the WMO scale but have not yet done so. Like the reviewer, we are not aware of an external scale for Ne.

Additional details regarding preparation and analysis of standards have been added to the text on lines 188-206.

Line 179: What were the resulting concentrations in the working standards? How many working standards did you produce and use? Same question for Line 238 with the part of the sentence saying "several concentrations". Please specify at least in one of these text passages.

Revised text (Line 207): "The weekly preparation of the working standards ensured that the 2L flask wouldn't show any  $H_2$  drift over the course of the analysis. Prior laboratory measurements showed that the 2L flasks would drift over the course of 3-4 weeks. Four sets of field working standards (about 25 total) were prepared over the course of 1 month. The field working standards contained concentrations ranging from 150 to 500 ppb for  $H_2$  and 10 ppm to 25 ppm for Ne."

Line 180: Did you see a difference whether  $N_2$  or zero air is used as the balance gas, or does it not matter?

There is no difference between  $N_2$  and zero air in terms of response. Zero-air did exhibit higher levels of  $H_2$  in blanks upon arrival from the supplier (Airgas). We removed mention of zero air in the text, since only  $N_2$  was used in the field.

Line 183: Were working standards analyzed once or multiple times daily? Please specify in the text (or in Line 238). How was the necessary frequency of working standard analysis assessed?

Revised text line 207: "Each day, four different working standards were analyzed. The standards were run in duplicate or triplicate throughout the day for a total of 8-10 working standard runs each day."

The working standards were run before, between, and after ice core samples.

Line 183: I assume that the weekly preparation of working standards has to do with limitations at this remote site? If there was a method-related reason, please specify in the text.

Daily dilutions were not possible due to time limitations on the field. The weekly preparations fit better in these time constraints. Line 207 explains: "The weekly preparation of the working standards ensured that the 2L flask wouldn't show any  $H_2$  drift over the course of the analysis. Prior laboratory measurements showed that the 2L flasks would show  $H_2$  over the course of 3-4 weeks."

**Line 187: connected to the "vacuum line" (I presume)**

Revised caption for Fig. 5: "The low-pressure flask and high-pressure gas standard are connected to the vacuum line using PFA tubing."

Figure 4: Please also mention the calibrated volume briefly in the text – or is this meant by the stainless-steel vacuum line?

This text was revised for clarity in line 192: "The high-pressure cylinder standards are prepared by expanding the pure gases ( $H_2$ , Ne,  $CH_4$ ,CO) from a section of the stainless-steel vacuum line at known pressure, volume, and temperature into the evacuated stainless-steel cylinder.  $N_2$  is then added to the cylinder through a high-pressure transfer line. The added  $N_2$  is determined gravimetrically."

Line 198: What was the eventual temperature range that the aluminium frame box could create?

Revised text (Line 225): "Gas flows, GC oven temperature, and detector temperature were maintained at the normal operating values overnight. The nighttime power draw of the system was approximately 80W which provided heating for the insulated box. The in-line fan was activated in order to cool the box if the temperature in the box reached 20 °C, which did occur during the deployment."

Line 212: I understand that the GC and detector were always restarted anew before an analysis batch? The stabilization of a GC-PDD can take some time after restart. Did you observe this issue? In case yes, I assume the GC and the detector were already running some test runs before the actual sample analysis, to have the baseline stabilised?

See revised text at line 225 (above).

Line 252: I have already commented on this above, but could this daily variation of residuals be an artefact of restarting the instrument every day? Or is the instrument running continuously, or is it in idle over night?

See revised text at line 225 (above).

Line 218: The level of detector output/ baseline varies over the runs. Is this the drift you are addressing later in the text and that needed correcting? What is the reason for this, i.e. is there a connection with the operating time of the instrument, or parameters like temperature or pressure, or is it a random pattern?

The baseline did not actually vary from run to run. We revised the caption of Fig. 6 to explain that: "The detector signals are shifted vertically so as not to overlap.:

Line 240-259: I am not familiar with this procedure for calibrating measurements. I presume the purpose of all these steps is to, firstly, establish a calibration curve and secondly, to apply a drift correction. Please specify this at the start of the description of the steps, e.g. in Line 239. Also, if possible, please reference to respective literature for this data treatment method.

That is correct. The correction process was unusually complicated for this project due to the harsh conditions and sensitivity changes caused by a faulty electronic pressure controller. Under normal operating conditions, more typical chromatography calibration procedures would suffice. There is no literature that we are aware of to cite for the data treatment we used.

Line 245: If injection loop temperature should actually also be accounted for, why was it not included, even if it did not improve or even worsened the regression statistics? If including it did not have a large effect on the statistics, then it could still have been included, or the sentence should at least be rephrased saying "did not change the regression statistics" or similar.

Revised text line 276: "In theory, injection loop temperature should also be accounted for in the regression, but including it did not improve the regression statistics."

Line 8: the abbreviation needs to be "He-PDD", not "He-PPD"

Done

Line 101: a "to" is missing, "due to compressive"

Done

Line 103: a "the" is missing, "because the O-ring"

Done

Line 192: a "to" is missing, "due to"

Done

Line 204: an "of" is missing, "of the ice"

Done

Line 246: I think "Ne" has to be "neon", as this is used throughout the text and to make the naming consistent; please check for consistent naming throughout the manuscript.

Done

Line 248: I cannot find section 2.1.3, please correct the number in the cross-reference.

Done

Figure 7: Maybe I missed it, but what are the red circles in the plot? Please add description to caption.

Done

Figure 7: I think the depth range "75-150 m" in the caption does not match the x-axis of the plot

Done

Figure 7: The gas age range "150-1750" in the caption does not match the age range in Line 271.

Done